# Effects of Different Lactic Acid Bacteria in Single or Mixed Form on the Fermentative Parameters and Nutrient Contents of Early Heading Triticale Silage for Livestock

**DOI:** 10.3390/foods12234296

**Published:** 2023-11-28

**Authors:** Ilavenil Soundharrajan, Jeong Sung Jung, Karnan Muthusamy, Bae Hun Lee, Hyung Soo Park, Ravikumar Sivanesan, Ki Choon Choi

**Affiliations:** 1Grassland and Forages Division, National Institute of Animal Science, Rural Development Administration, Cheonan 31000, Republic of Korea; ilavenil@korea.kr (I.S.); jjs3873@korea.kr (J.S.J.); karnantm111@gmail.com (K.M.); leebaehun@korea.kr (B.H.L.); anpark69@korea.kr (H.S.P.); 2Department of Zoology, Rajah Serfoji Government College (Autonomous), Thanjavur 613-005, India; drravinikesh@yahoo.co.in

**Keywords:** LAB, triticale, anaerobic fermentation, silage quality, organic acid

## Abstract

Lactic acid bacteria (LAB) are excellent anaerobic fermenters that produce highly valuable grass-based animal feed containing essential nutrients. In the present study, an ensiling process was used to improve anaerobic fermentation in triticale silage under different moisture conditions with LAB. The triticale was treated with either a single bacterium or combined LAB and then vacuum-sealed. After 180 and 360 days of storage, the silage’s fermentation characteristics, microbial changes and nutrient contents were analyzed. The pH of the silage was significantly lower than the control silage. There was a significant difference in the pH values between the silages treated with single or mixed LAB. The LAB treatment led to a substantial increase in lactic acid (LA), a decrease in butyric acid (BA), and marginal levels of acetic acid (AA). The LA content after the mixed LAB treatment was significantly higher than that after the single culture LAB treatment. After single or combined inoculant treatments, the LAB population in the silage increased, while the yeast and mold levels decreased. These findings suggest that the addition of LAB to silage during ensiling could enhance the nutritional quality and reduce unwanted microbial growth. The mixed LAB treatments produced silage with a significantly higher nutritional value than the single LAB treatments.

## 1. Introduction

As animal husbandry has developed rapidly, shortages of high-quality feed have increased; therefore, it is very important to develop new high-quality feeds from grasses and legumes. A triticale crop was developed in German laboratories from wheat and rye in the 19th century [1], and the triticale species has been used based on its characteristics. The chemical composition of most of these species is similar to that of wheat, which is used in human and animal nutrition. The higher crude protein content makes it useful for ruminant feed production [2,3]. Currently, only a few studies have been conducted using triticale with LAB for feed development [3,4,5]. The benefits of triticale include a higher biomass, faster growth in the spring and longer mowing times [2].

In many parts of the world, preserved forages, such as silages, are used as a major part of a livestock’s diet. Under anaerobic conditions, native lactic acid bacteria (LAB) on plant surfaces play an important role in lowering the pH of silage during natural ensiling, inhibiting undesirable microbes, decreasing feedstock degradation risk, and preserving forages for a long time [3,6]. In silage production from grasses, legumes, and other plants, lactic acid bacteria (LAB) play a critical role. Since ensiling is a consistent, reliable, and predictable method for producing feed, it is becoming increasingly popular. Various factors may reduce the conservation efficiency, including plant oxidation, microbial population, proteolytic activity, Clostridia fermentation, and microbial deamination and decarboxylation. Thus, silage samples accumulate anti-nutritional compounds and lose energy and nutrients. During ensiling, LAB produce different types of organic acids, including lactic acid, acetic acid, butyric acid, succinic acid, etc. In silage, acidification inhibits undesirable microbial growth and prevents aerobic degradation [7,8]. In addition to affecting the silage’s nutritional quality, spoilage microorganisms can also affect animal health and animal products. The use of LAB has long been implemented to improve the quality of low-/high-moisture silage around the world, as well as to increase milk production, body weight, and feed intake efficiency [8,9]. Under anaerobic conditions, *Lactobacillus*, *Pediococcus*, and *Enterococcus* are the major genera involved in silage fermentation. In particular, *Lactiplantibacillus plantarum*, *Levilactobacillus brevis*, *Pediococcus pentosaceus*, *Lacticaseibacillus rhamnosus*, *Lentilactobacillus buchneri*, and *Enterococcus faecium* are the most common LAB species intensively used as inoculants for silage production [3,10,11]. Forage material that has been ensiled is exposed to the air when the silo is opened and thus begins to deteriorate, particularly lactate-assimilating yeast, which leads to an increase in the silage temperature and a rise in pH, which favor undesirable microbial growth [12]. To address this issue, lactic acid bacteria have been used as inoculants in silage for several decades [12,13]. The limited amount of LAB in native plants makes it difficult to produce high-quality silage via direct ensiling. It has, therefore, been common to isolate novel LAB for silage production in recent decades, and this activity continues to be necessary around the world today [14,15] since, increasingly, more strains are being sought for future use, not only as silage inoculants, but also in other plant-based foods for animals and humans [16]. The majority of studies have found that LAB strains promote silage fermentation as a single culture [8,17]. However, interestingly, during silage production with mixed cultures of LAB (more than one strain), the acidification of silage was accelerated, the lactic acid production was enhanced, and butyric acid was reduced via synergistic effects compared to monoculture LAB treatment [13,18,19]. Taking these facts into account, we studied the effects of different types of lactic acid bacteria on acidification, fermentative metabolites, the microbial population, and nutrients of fermented triticale silage for different storage durations and moisture levels.

## 2. Materials and Methods

### 2.1. Inoculant Preparation for Silage Production

*L. plantarum-KCC-34* [20], *L. plantarum-KCC-48* [21], *P. pentosaceus-KCC-44* [22], *P. pentosaceus-KCC-53* (*GenBank* accession No. MZ505239), and *L. rhamnosus–KCC-54* (*GenBank* accession No. MZ505240) strains were isolated from different sources and have been reported previously. All the strains were cultured in MRS broth (CONDA, Madrid, Spain) for 30 h with mild shaking at 150 rpm in an orbital shaker under micro-aerobic conditions at 37 °C, and the pellets were removed by centrifugation at 4000× *g* for 45 min at 4 °C. After washing twice with phosphate-buffered saline (PBS, pH 7.4), sterile distilled water was used to dilute the bacterial pellets. A Quantom Tx Microbial cell counter (Logos Bio-system, Anyang-si, Republic of Korea) was used to count the bacterial colonies. One microliter of Quantom total cell staining enhancer was mixed well with ten microliters of the diluted sample, then left at room temperature for 30 min. The mixtures were thoroughly mixed, without the introduction of bubbles, with eight microliters of cell loading buffer. The prepared samples (6 μL) were loaded onto Quantom M50 cell counting slides and centrifuged for 10 min at 300× *g*. Quantom cell counters were used to count the bacteria, and sterile water was used to dilute the bacteria to 10^5^ cells/mL.

### 2.2. Plant Collection and Silage Production

The Joseong cultivar triticale was harvested early in Jangsoo, Chunbuk (latitude: 35.6185318, longitude: 127.5107881), Republic of Korea, and allowed to wilt for 8–10 h, or 24–36 h for low-moisture silage [3]. A manual cutter was used to chop 300 g of the whole crop of triticale into 1.5–2.5 cm pieces. The chopped samples were placed in 28 × 36 cm silage bags (Aostar Co., Ltd., Seoul, Republic of Korea). Each moisture sample bag was divided into eight groups with three replicas (*n* = 3) of each group. The experimental groups were Group-I (non-inoculant), Group-II (*L. plantarum*-KCC-34), Group-III (*P. pentosaceus*-KCC-44), and Group-IV (*L. plantarum*-KCC-48), as well as Group-V (*P. pentosaceus*-KCC-53), Group-VI (*L. rhamnosus*-KCC-54), Group-VII (KCC-44 + 48 + 53), and Group-VIII (KCC-34 + 44 + 54). The bags were vacuum-sealed (Food Saver V48802, MK Corporation, Seoul, Republic of Korea) to remove air. The bags were stored under laboratory conditions for 180 and 360 days for each group.

### 2.3. Sampling and Analysis of Fermentative Metabolites

To analyze the fermentation characteristics, 10 g of each sample was mixed with 90 mL of water and shaken in an orbital shaker for an hour. A glass electrode pH meter (Thomas Scientific, Swedesboro, NJ, USA) was used to measure the pH of the filtrate after it had been passed through multiple layers of cheesecloth and a 0.2 μm filter membrane. Afterwards, the organic acid concentrations in the fermented samples were determined. The samples were reduced to a pH of 2 with 50% sulfuric acid and frozen at −20 °C for HPLC analyses. The lactic acid concentration in the experimental silage was determined using a high-performance liquid chromatography system with a G1321A FLD detector (HP1100, Agilent Co., Santa Clara, CA, USA). A Hi-Plex Ligand exchange column (300 × 7.7 mm) from Agilent was used to elute the sample at 40 °C with 0.1 M H_2_SO_4_. An HPLC analysis was conducted at a flow rate of 0.6 mL/minute and a wavelength of 220 nm. A CP7485 column fused with silica (length: 25 cm; diameter: 0.32; film thickness: 0.30) and with temperature ranges from 20 °C to 270 °C was used to analyze the acetic and butyric acid content in the silage. The sample flow rate was 10 microliters per minute [3,23].

### 2.4. Analysis of Nutrient Contents

The initial weight of each sample was determined, and then the samples were dried at 60 °C in an oven. The dry matter content (DM) was immediately determined for each sample, and the samples were then ground in a cutting mill and stored until further use. In order to determine the crude protein content in the samples, the Kjeldahl method [24] was used. The ADF (acid detergent fiber) and NDF (neutral detergent fiber) contents were quantified [25].

### 2.5. Microbial Enumeration in Experimental Sample 

For the enumeration of the LAB, molds, and yeast, a portion of the sample was filtered with sterilized cheesecloth and then serially diluted tenfold in sterile distilled water. Then, 0.1 mL of each sample was poured on a de Man–Rogosa–Sharpe agar plate (MRS agar, CONDA, Madrid, Spain) and incubated at 37 °C for 48 h under an aerobic condition. Further, 1 mL of the diluted sample was spread on petrifilm for the detection of molds and yeast (3M microbiology products, St. Paul, MN, USA) and incubated at 37 °C for 70 to 120 h. After the respective incubation periods, the populations of the microbes were enumerated.

### 2.6. Statistical Analysis

A randomized design was used with eight treatments and three replicates per treatment. A least significant difference test was used to analyze the significant differences using SPSS16 software (one-way ANOVA, multivariate analysis, post hoc, Duncan, and descriptive analysis parameters). Statistical significance was determined by a *p*-value of less than 0.05.

## 3. Results

### 3.1. Impacts of LAB Strains on Nutrient Contents

The nutrient contents of the control and LAB-treated triticale silage were determined at different moisture levels after 180 and 360 days. The high-moisture silage treated with monoculture *L. plantarum*-KCC-48 (ADF: 22.7 ± 0.69% DM) and mixed LAB KCC-34 + 44 + 54 (ADF: 23.3 ± 0.91% DM) had significantly reduced ADF contents compared to the control (ADF:26.7 ± 0.09% DM) after both storage periods (*p* < 0.05). The NDF and CP contents were not significantly affected by any of the LAB treatments, moisture levels, or storage periods. A slight reduction in NDF and an increase in the CP level were observed in the HM silage treated with *L. plantarum*-KCC-48 after all the storage periods (Table 1 and Table 2).

### 3.2. Reduction in pH of Experimental Silages by LAB Treatments

The pH values of the experimental silages in response to different treatments, storage periods, and moisture levels are shown in Figure 1a and Figure 2a. The silage under both the high-moisture (HM) and low-moisture (LM) conditions without inoculants exhibited higher pH values on day 180 (HM: pH 5.23 ± 0.01 and LM: 6.12 ± 0.04) and day 360 (pH 5.14 ± 0.30 and 5.49 ± 0.56). In contrast, the silage treated with different LAB as a single culture exhibited reduced pH values: the pH of the high-moisture triticale silage ranged between 3.89 ± 0.05 and 4.15 ± 0.24 and that of the low-moisture triticale silage ranged between 4.31 ± 0.05 and 4.85 ± 0.05 on day 180. Similarly, the triticale silage produced with different LAB strains showed reduced pH values under high-moisture (ranging between pH 3.99 ± 0.12 and 4.73 ± 0.07) and low-moisture (ranging between pH 4.21 ± 0.12 and 4.43 ± 0.23) conditions on day 360 compared to the control. The silage treated with mixed LAB exhibited a strong reduction in the pH; however, there were no significant changes in the pH of mixed LAB-treated silage compared to the single culture treatments for both moisture levels and storage periods.

### 3.3. Fermentative Metabolite Production in Silage for Different Moisture Levels and Storage Periods

The lactic acid (LA), acetic acid (AA), and butyric acid (BA) contents were determined in the experimental silages on days 180 and 360 and are shown in Figure 1b–d and Figure 2b–d. The LA content of the control silage was 1.57 ± 0.12 and 1.23 ± 0.18% DM in HM conditions and 0.04 ± 0.00 and 0.63 ± 0.22% DM in LM conditions, respectively, after 180 and 360 days. On day 180, the LA content in the silage treated with different single culture LAB increased sharply compared to the control silage. On day 180, the LA levels in the HM and LM silage treated with LAB ranged from 3.75 ± 0.08 to 7.37 ± 0.14% DM and 2.02 ± 0.02 to 4.86 ± 0.48% DM, respectively. The KCC-48 treatment led to a higher LA content in the silage than the other individual strains or compared to the control silage. When the silage was treated with mixed LAB strains, the LA production accelerated. In particular, the KCC-34 + 44 + 54 combination produced more LA in the silage on day 180 than the other combination (KCC-44 + 48 + 53) in HM conditions; however, the co-culture of KCC-44 + 48 + 53 led to a higher LA content in the LM silage than the co-culture of KCC-34 + 44 + 54. On day 360, the silage treated with different LAB strains as a single culture showed a wide range of LA values, from 2.82 ± 0.21 to 3.12 ± 0.22% DM in HM conditions and from 3.16 ± 0.39 to 5.28 ± 0.46% DM in LM conditions.

The mixed LAB strains accelerated the LA production in the HM silage, especially KCC-34 + 44 + 54, compared with KCC-44 + 48 + 53 on day 360. The silage treated with KCC-44 + 48 + 53 or KCC-34 + 44 + 54 in LM conditions exhibited no significant change in LA content on day 360. Despite this, KCC-48 showed strong LA production in the silage for both moisture levels and both storage durations. In the experimental silages, the acetic acid levels varied with different treatments and storage periods. In the high-moisture silage on day 180, the AA levels in the fermented silage were significantly reduced by KCC-53 and KCC-54, while the silage treated with KCC-48 alone and mixed strain KCC-34 + 44 + 54 had significantly increased AA levels. The rest of the LAB treatments had no effect on the silage’s AA contents. In the LM silage, the individual KCC-44 and KCC-48 treatments increased the AA levels compared to the control silage, whereas the mixed LAB treatment reduced the AA levels compared to the KCC-44 and KCC-48 treatments. The AA content of the silage was not affected by KCC-53 or KCC-54 alone in comparison to the control (Figure 1b–d). The single or mixed LAB-treated silage had increased AA levels on day 360, except for treatment with KCC-54 and KCC-34 strains alone under high-moisture conditions. The LM silage had higher AA levels in the control than the LAB-treated silage. In contrast, the AA levels in the silage treated with single or mixed LAB were significantly reduced (Figure 2b–d). Different treatments also resulted in different levels of butyric acid (BA). The control HM silage had higher BA levels (0.15 + 0.05), but the LAB treatments, either alone or mixed, prevented BA production. Different LAB treatments resulted in lower BA levels in the HM silages compared to the control (*p* < 0.05). On days 180 and 360, there were no significant changes in BA production in the LM silages when different treatments were applied. 

### 3.4. Microbial Changes in Experimental Silage for Different Storage Periods and LAB Treatments

The microbial populations, such as the total lactic acid bacteria (LAB), yeast, and mold, were counted in both the HM and LM silages on day 180 in response to different LAB treatments (Figure 3a–c). Under high-moisture conditions, the number of LAB was 11.5 × 6.3 × 10^7^ CFU/g, while under low-moisture conditions, it was 5.0 ± 1.4 × 10^7^ CFU/g in the control. The yeast and mold counts for the control HM silage were 341 ± 1.55 × 10^5^ CFU/g and 20.5 ± 6.36 × 10^4^ CFU/g, and for the LM silage, they were 24.3 ± 0.40 × 10^5^ CFU/g and 12.2 ± 1.25 × 10^4^ CFU/g, respectively. The silage treated with different types of LAB, either as a single or co-culture, had higher LAB populations than the control silage. The LAB numbers in the HM silage varied with the inoculant treatments, ranging from 30.5 ± 4.9 to 81.0 ± 4.2 × 10^7^ CFU/g. The yeast and mold counts were also drastically reduced or insignificant when the silage was treated with different LAB monocultures or co-cultures.

Figure 4a–c shows the microbial profiles of LAB, yeast, and mold in the control and LAB-treated triticale silage on day 360. The control group contained 9.5 ± 2.1 × 10^7^ CFU/g of LAB in the HM silage and 3.5 ± 1.4 × 10^7^ CFU/g in the LM silage. In both the HM and LM conditions, the yeast and mold numbers were higher in the silage produced without different LAB. The LAB treatment in a single or mixed culture significantly either inhibited or reduced both the yeast and mold counts. The total LAB ranged from 23.5 ± 7.7 × 10^7^ CFU/g to 65.5 ± 4.9 × 10^7^ CFU/g in the silage treated with different inoculants in HM conditions. A LAB population ranging between 3.5 ± 0.7 and 50 ± 13 × 10^7^ CFU/g was found in the silage treated under low-moisture conditions with different inoculants. In addition, the yeast and mold counts were drastically reduced or prevented when the silage was treated with different LAB monocultures or co-cultures.

## 4. Discussion

A high silage fermentation quality is always accompanied by nutrient preservation. In order to preserve roughages, ensiling is a method that has been used for decades to spontaneously produce lactic acid under controlled fermentation conditions in anaerobic environments. Forage preservation by ensiling has gained significant attention for providing consistent, reliable, and predictable feed supplies for ruminants. Plant oxidation, undesirable microbial populations in plants, proteolytic activity, *Clostridia* fermentation, microbial deamination, and decarboxylation of amino acids could negatively affect efficiency and result in higher energy, nutrient losses, and anti-nutritional compound accumulation in forage samples [26]. By utilizing the water-soluble carbohydrates present in ensiled plants, epiphytic lactic acid bacteria produce lactic acid and a lower amount of acetic acid, which lowers the pH of the silage, preventing undesirable microorganisms from growing and allowing the silage to be stored for a long time. The abundance of epiphytic bacteria in ensiled plant materials is not sufficient to induce lactic acid production in silage samples. Plant populations of LAB are often heterofermentative and low in number [27]. To make high-quality silage with high digestibility, different types of additives need to be included in the ensiling process. LAB are used as an inoculant to increase the ratio of lactic acid to acetic acid in silage production, to reduce proteolysis, and to increase dry matter recovery [28,29]. According to several studies, adding LAB to silage produced from different forages could promote positive fermentation [30,31]. Meanwhile, ensiling high-quality silages with mixed LAB strains rapidly accelerates silage fermentation due to their synergistic effects of increasing the LA content and lowering the BA content compared to monoculture-based silage production. There is evidence that the LAB treatments performed in mixed form produce higher LA and significantly reduce unwanted microbes in comparison with single strain treatments. In addition, a LAB co-culture treatment extends the fermented silage storage time while preserving nutrients [13,18,19,32]. Considering the observations above, whole-crop triticale silage was produced under different moisture conditions in the presence of different LAB strains, using either single or co-culture treatments, for 180 and 360 days. On days 180 and 360, the fermentative parameters, nutrient contents, and microbial profiles of the fermented silages were evaluated. Silage is enriched with nutrients that maintain the fermentation quality. ADF and NDF contents are key indicators of silage quality [33,34,35]. Increased ADF, NDF, and lignin contents in the silage indicate poor silage quality and reduce its digestibility, which is not beneficial for animal digestion. Fiber-rich silages contain less protein and energy than silages with lower fiber contents; thus, reducing the fiber content is a good strategy to improve the feed value [36]. According to the results of this study, treating silage with most of the LAB under different moisture conditions had no significant impact on the nutrient contents, such as the ADF, NAD, and CP levels. In contrast, KCC-48 and mixed KCC-34 + 44 + 54 significantly (*p* < 0.05) reduced the ADF content in HM silage on days 180 and 360. However, the same LAB treatment did not affect the ADF, NDF, or CP contents of the LM silage. There is evidence to suggest that the silage’s moisture content has an impact on the LAB activity regarding ADF degradation. LAB growth and colonization require optimum moisture conditions to promote fermentation. On days 180 and 360, KCC-48 alone slightly reduced the NDF contents and increased the CP levels in the silage. The silage treated with KCC-48 along with other strains as a co-culture did not exhibit any changes in NDF and CP contents, suggesting that KCC-48’s activity might be lowered when mixed with other LAB.

In order to produce high-quality silage, pH is an essential factor. When the pH is rapidly reduced in silage, the proteolytic enzyme activity is reduced, preventing nutrient decomposition and inhibiting the growth of unwanted bacteria, including *enterobacteria* and *clostridia* [7]. The ideal pH for silage fermentation is between 3.8 and 4.2 [37]. A pH of 4.2 is also considered a benchmark for well-conserved silage, particularly for high-moisture silage [33]. The pH of the control silage produced without inoculants was higher than those produced with inoculants for all the moisture conditions and storage periods, indicating inadequate fermentation occurred. It may be that inadequate LAB numbers are present in the plant samples, causing poor silage fermentation. This is consistent with the low LAB population and high yeast and mold populations in the triticale silage produced without inoculants. However, the silage treated with different LAB strains exhibited a sharp reduction in pH compared to the control silage (*p* < 0.05). For both moisture levels and both storage periods, the pH almost reached the desirable level in response to different LAB treatments. A reduced silage pH is caused by a higher LAB population and lower mold and yeast counts. It was found that the silages treated with LAB, whether single or co-cultured, showed significant strain-dependent variations (*p* < 0.05) in pH reductions.

LAB are usually used in the production of high-quality silage since they produce higher levels of lactic acid than other organic acids from water-soluble carbohydrates [38]. In general, lactic acid is the dominant acid accumulated in silage during microbial fermentation. Its levels are approximately 10–12 times greater, and it contributes to lowering the pH of silage [12]. The results of the LAB treatment were a higher lactic acid concentration, a lower pH, and a lower butyric acid concentration, suggesting that the fermentation pattern in the LAB-treated triticale silage was homofermentative. This can be confirmed by the homofermentative properties of the LAB used in this experiment. The high-moisture silage treated with different LAB produced different levels of lactic acid. In the control silage, the amount of lactic acid was lower, which indicates that LAB in native plants do not produce enough lactic acid for positive silage fermentation. However, the silages produced with different LAB produced more lactic acid than the control silages by more than two- to four-fold. The mixed LAB treatments further accelerated the LA production compared to the single LAB treatments, with the exception of *L. plantarum*-KCC-48. Similar LA production trends were observed in the LM silages in response to different LAB treatments compared to the control silages. There was a slight reduction in the LA percentage when compared to the HM silage because of the forage’s low moisture content. In order for LAB to grow and colonize, water-soluble carbohydrates (WSCs) are necessary. When the moisture is low, WSC availability is low, which affects LAB growth and LA production, resulting in a large variation in the LAB population between the HM and LM silages. Even after extended storage periods of 360 days, LA concentrations of more than 2% were found in the silage treated with different LAB strains. The percentage of LA in the silage was slightly lower on day 180 compared to the HM silage on day 360. It has been suggested that the LA content decreases during ensiling because of the lower availability of WSCs in the silage, leading to slow LAB fermentation [39]. Furthermore, some anaerobic microbes present in the samples could decompose LA into propionic acid, which could result in a reduction in LA levels [40]. In addition, yeast degrades LA into ethanol [41]. The current study confirms the statements above by showing that the LA content in the HM silage treated with different LAB on day 360 was lower compared to day 180, possibly as a result of higher yeast counts and other anaerobic bacterial growth. In contrast, the LA content of the LM silage is inconsistent with that determined by Shao et al., despite there being a higher yeast population on day 360 than on day 180, yet the LA content was higher in the LM silage on day 180. A prolonged storage period may soften forages, affecting the WSC contents in plants and their continuous availability for constant LAB growth and prolonged LA production. Even after prolonged storage times (360 days), the silage produced with different types of LAB in single or mixed form exhibited typical fermentation characteristics with a significant lactic acid content. This finding is consistent with that of Kleinschmit and Kung et al., who reported that corn silage produced with *L. buchneri* and *P. pentosaceus* exhibits normal fermented silage characteristics even after 361 days [18]. Based on these findings, it is concluded that LA production and maintenance in fermented silage are also sustained for a long time.

After lactic acid, acetic acid is the second most dominant acid in fermented silage [12,42]. It inhibits yeast and therefore improves the silage’s aerobic stability. In silage, maintaining a certain amount of AA is desirable in order to reduce the growth of yeast and mold during aerobic exposure [43]. The AA range is often recommended to be between 10 and 30 g/kg DM [12,44]. The HM silage treated with KCC-48 and mixed LAB KCC-34 + 44 + 54 had significantly higher AA contents than the control silage. A higher amount of AA was observed in the KCC-44-, KCC-48-, KCC-44 + 48 + 53-, and KCC-34 + 44 + 54-treated LM silages than in the control silages. In the HM silage, on day 360, the AA content significantly increased in response to KCC-44, KCC-48, KCC-53, and mixed LAB. Both the HM and LM silages produced an acceptable amount of AA; their ranges are within acceptable limits. All the LAB strains inhibited AA production in the LM triticale silages on day 360 compared to the controls. In addition, in the silage treated with different LAB, the butyric acid production was strongly inhibited, a negative indicator of silage quality [45], suggesting that the inoculants used improved the triticale silage quality by increasing the LA content with marginal AA levels and inhibiting BA production. The effects of all the strains on the triticale silage fermentation parameters differed significantly, but the mixed LAB synergistically significantly enhanced the silage quality.

## 5. Conclusions

A study was conducted to evaluate the effects of different LAB strains, namely, *L. plantarum-KCC-34*, *L. plantarum-KCC-48*, *P. pentosaceus-KCC-44*, *P. pentosaceus-KCC-53*, and *L. rhamnosus-KCC-54*, on the fermentation parameters, nutrient contents, and microbial population of early heading triticale silage with different moisture levels after ensiling for 180 and 360 days. The silage produced with different LAB, either as a single bacterium or mixed LAB, significantly reduced the pH and butyric contents and increased lactic acid contents by inhibiting the growth of yeast, mold, and other unwanted bacteria. Triticale silage fermentation was influenced by different LAB strains to varying degrees. Treatments with LAB preserved significant amounts of lactic acid and the low pH in triticale silage even after prolonged storage. All the single LAB treatments significantly improved the silage quality in different ways; however, *L. plantarum*-KCC-48 had a more significant effect on silage fermentation and preservation. Furthermore, adding mixed LAB sped up the fermentation of triticale silage via synergistic effects. These findings suggest that the early heading silage fermentation quality is affected by the form of inoculants, the moisture conditions, and the storage duration.

## Figures and Tables

**Figure 1 foods-12-04296-f001:**
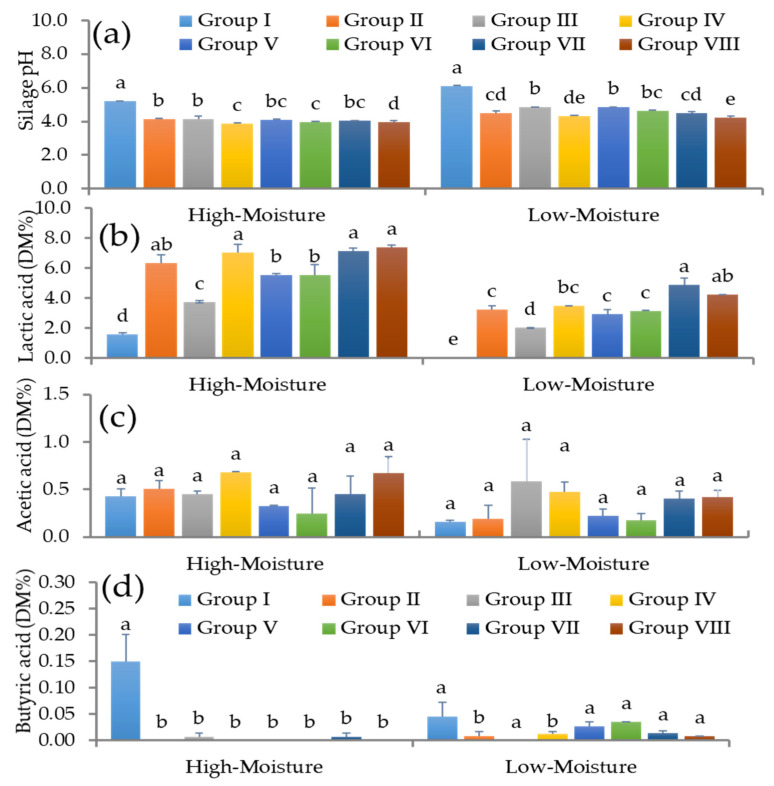
The organic acid profile of early heading triticale silages treated with single and co-culture LABs at different moisture levels on day 180. (**a**) pH values of experimental silages; (**b**) lactic acid content of control and LAB-treated silages; (**c**) acetic content of control and LAB-treated silages; (**d**) butyric acid content of experimental silages. I—Control; *II*—*L. plantarum*-KCC-34; III—*P. pentosaceus*-KCC-44; IV—*L. plantarum*-KCC-48; V—*P. pentosaceus*-KCC-53; VI—*L. rhamnosus*–KCC-54; VII—KCC-44 + 48 + 53; VIII—KCC-34 + 44 + 54; DM—dry matter content. Different letters within a column indicate a significant differences between treatments and control silages (*p* < 0.05).

**Figure 2 foods-12-04296-f002:**
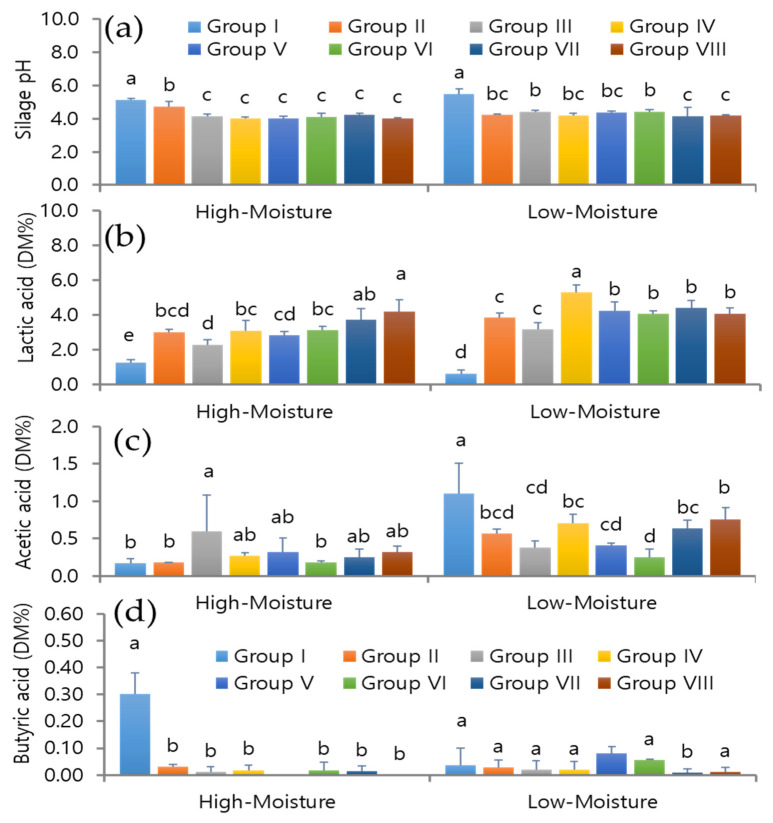
Analysis of organic acid profiles in early heading triticale silages treated with single or co-cultured LAB under different moisture conditions on day 180. (**a**) pH values of experimental silages; (**b**) lactic acid content of control and LAB-treated silages; (**c**) acetic content of control and LAB-treated silages; (**d**) butyric acid content of experimental silages. I—Control; *II*—*L. plantarum*-KCC-34; III—*P. pentosaceus*-KCC-44; IV—*L. plantarum*-KCC-48; V—*P. pentosaceus*-KCC-53; VI—*L. rhamnosus*-KCC-54; VII—KCC-44 + 48 + 53; VIII—KCC-34 + 44 + 54; DM—dry matter content. Different letters within a column indicate significant differences between treatments and control silages (*p* < 0.05).

**Figure 3 foods-12-04296-f003:**
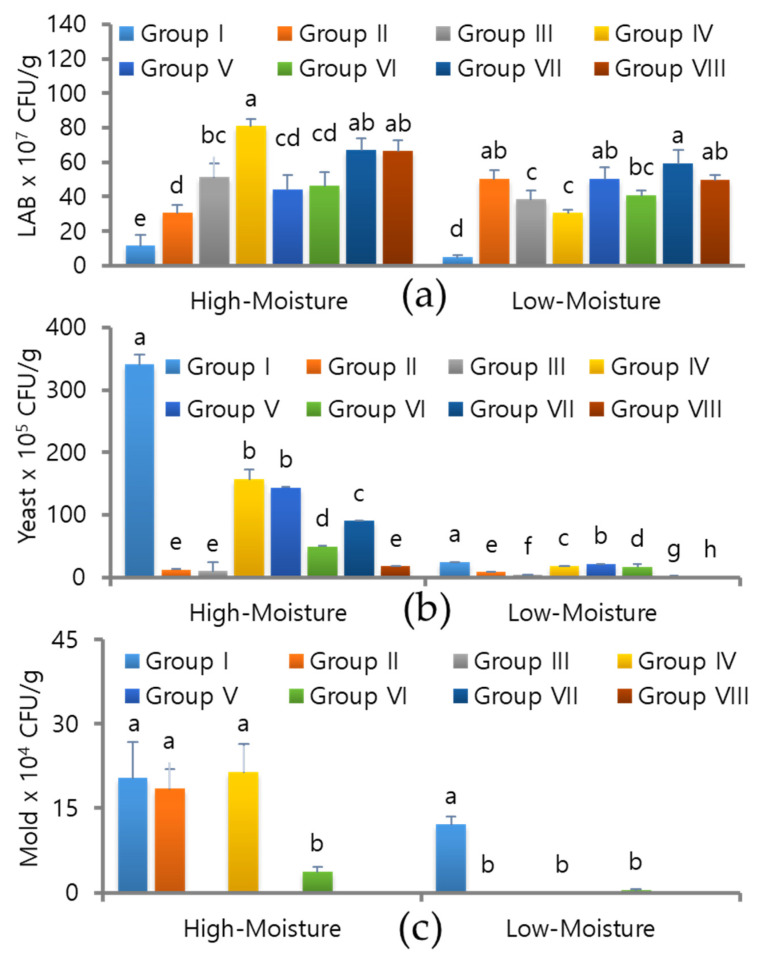
Microbial changes in early heading triticale silage under different moisture conditions in response to single or co-culture LAB treatments on day 180. (**a**) Lactic acid bacterial profiles in control and LAB-inoculated silages on day 180; (**b**) yeast counts in control and LAB-inoculated silages on day 180; (**c**) mold population in control and LAB-inoculated silages on day 180. LAB × 10^7^ CFU/g: lactic acid bacteria/colony forming unit/gram; I—control; II—*L. plantarum*-KCC-34; III—*P. pentosaceus*-KCC-44; IV—*L. plantarum*-KCC-48; V—*P. pentosaceus*-KCC-53; VI—*L. rhamnosus*-KCC-54; VII—KCC-44 + 48 + 53; VIII—KCC-34 + 44 + 54. Different letters within a column indicate a significant difference between treatments (*p* < 0.05).

**Figure 4 foods-12-04296-f004:**
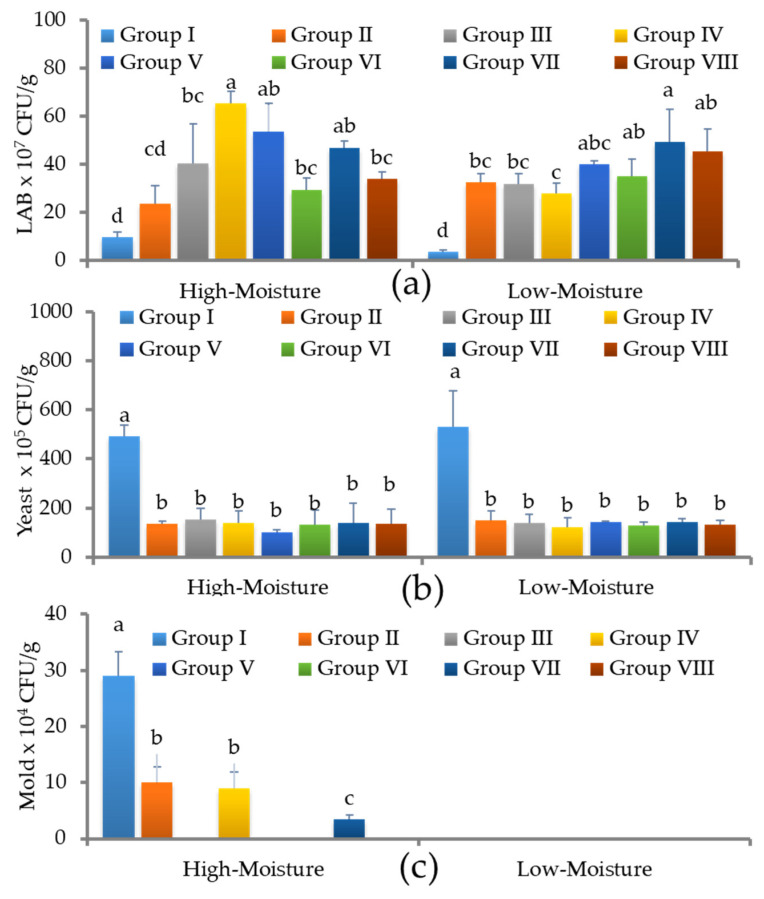
Microbial changes in early heading triticale silage under different moisture conditions in response to single or co-culture LAB treatments on day 360. (**a**) Lactic acid bacterial profiles in control and LAB-inoculated silages on day 180; (**b**) yeast counts in control and LAB-inoculated silages on day 180; (**c**) mold population in control and LAB-inoculated silages on day 180. LAB × 10^7^ CFU/g: lactic acid bacteria/colony forming unit/gram; I—control; II—*L. plantarum*-KCC-34; III—*P. pentosaceus*-KCC-44; IV—*L. plantarum*-KCC-48; V—*P. pentosaceus*-KCC-53; VI—*L. rhamnosus*-KCC-54; VII—KCC-44 + 48 + 53; VIII—KCC-34 + 44 + 54. Different letters within a column indicate a significant difference between treatments (*p* < 0.05).

**Table 1 foods-12-04296-t001:** Early heading triticale silage nutrient profiles (%) under different moisture conditions after 180 days.

Groups	High Moisture			Low Moisture		
	ADF	NDF	CP	ADF	NDF	CP
Group I	26.7 ± 0.09 ^a^	47.6 ± 0.33 ^a^	23.0 ± 0.20 ^a^	26.3 ± 0.90 ^a^	47.6 ± 0.32 ^a^	22.8 ± 0.14 ^a^
Group II	26.0 ± 0.90 ^a^	47.3 ± 1.32 ^a^	23.1 ± 0.15 ^a^	26.1 ± 0.69 ^a^	47.5 ± 0.41 ^a^	23.4 ± 0.01 ^a^
Group III	27.1 ± 0.17 ^a^	48.1 ± 0.63 ^a^	23.1 ± 0.31 ^a^	26.9 ± 0.15 ^a^	48.0 ± 0.45 ^a^	23.2 ± 0.12 ^a^
Group IV	22.7 ± 0.69 ^b^	47.0 ± 0.41 ^a^	23.3 ± 0.01 ^a^	26.6 ± 0.42 ^a^	47.7 ± 0.26 ^a^	22.8 ± 0.05 ^a^
Group V	26.5 ± 0.15 ^a^	47.6 ± 0.45 ^a^	23.1 ± 0.13 ^a^	26.6 ± 0.74 ^a^	47.6 ± 0.55 ^a^	23.2 ± 0.34 ^a^
Group VI	26.3 ± 0.42 ^a^	47.5 ± 0.26 ^a^	23.1 ± 0.05 ^a^	27.1 ± 0.09 ^a^	48.0 ± 0.33 ^a^	23.4 ± 0.19 ^a^
Group VII	26.5 ± 0.29 ^a^	47.6 ± 0.54 ^a^	23.0 ± 0.09 ^a^	26.7 ± 0.42 ^a^	47.8 ± 0.26 ^a^	23.2 ± 0.05 ^a^
Group VIII	23.3 ± 0.9 ^b^	47.6 ± 0.45 ^a^	22.9 ± 0.11 ^a^	26.4 ± 0.90 ^a^	47.7 ± 0.32 ^a^	23.2 ± 0.14 ^a^

I—Control; II—*L. plantarum*-KCC-34; III—*P. pentosaceus*-KCC-44; IV—*L. plantarum*-KCC-48; V—*P. pentosaceus*-KCC-53; VI—*L. rhamnosus*-KCC-54; VII—KCC-44 + 48 + 53; VIII—KCC-34 + 44 + 54; ADF—acid detergent fiber; NDF—neutral detergent fiber; CP—crude protein. Different letters within a column indicate a significant difference between treatments (*p* < 0.05).

**Table 2 foods-12-04296-t002:** Nutrient profiles (%) of early heading triticale silages under different moisture conditions after 360 days.

Groups	High Moisture			Low Moisture		
	ADF	NDF	CP	ADF	NDF	CP
Group I	27.4 ± 0.09 ^a^	47.4 ± 0.33 ^a^	22.7 ± 0.19 ^a^	25.7 ± 0.87 ^a^	46.0 ± 1.23 ^a^	22.1 ± 0.14 ^a^
Group II	26.6 ± 0.90 ^a^	47.2 ± 1.31 ^a^	22.8 ± 0.14 ^a^	26.2 ± 0.12 ^a^	46.3 ± 0.45 ^a^	22.5 ± 0.09 ^a^
Group III	27.1 ± 0.12 ^a^	47.4 ± 0.43 ^a^	22.8 ± 0.14 ^a^	25.9 ± 0.70 ^a^	46.0 ± 0.50 ^a^	22.5 ± 0.34 ^a^
Group IV	23.7 ± 2.91 ^b^	46.9 ± 0.41 ^a^	23.0 ± 0.05 ^a^	25.9 ± 0.39 ^a^	46.1 ± 0.28 ^a^	22.1 ± 0.05 ^a^
Group V	27.0 ± 0.40 ^a^	47.3 ± 0.24 ^a^	22.7 ± 0.05 ^a^	26.5 ± 0.09 ^a^	46.4 ± 0.33 ^a^	22.6 ± 0.19 ^a^
Group VI	27.8 ± 0.17 ^a^	47.9 ± 0.61 ^a^	22.8 ± 0.29 ^a^	22.7 ± 4.41 ^a^	45.9 ± 0.37 ^a^	22.6 ± 0.00 ^a^
Group VII	27.1 ± 0.26 ^a^	47.4 ± 0.56 ^a^	22.8 ± 0.09 ^a^	25.8 ± 0.83 ^a^	46.1 ± 1.27 ^a^	22.5 ± 0.14 ^a^
Group VIII	23.9 ± 4.91 ^b^	47.4 ± 0.48 ^a^	22.6 ± 0.10 ^a^	26.1 ± 0.40 ^a^	46.2 ± 0.26 ^a^	22.4 ± 0.05 ^a^

I—Control; II—*L. plantarum*-KCC-34; III—*P. pentosaceus*-KCC-44; IV—*L. plantarum*-KCC-48; V—*P. pentosaceus*-KCC-53; VI—*L. rhamnosus*–KCC-54; VII—KCC-44 + 48 + 53; VIII—KCC-34 + 44 + 54; NDF—neutral detergent fiber; CP—crude protein. Different letters within a column indicate a significant difference between treatments (*p* < 0.05).

## Data Availability

This article contains all the experimented data.

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
