# Peer review of "Effects of Different Lactic Acid Bacteria in Single or Mixed Form on the Fermentative Parameters and Nutrient Contents of Early Heading Triticale Silage for Livestock"

_foods, 2023, doi:10.3390/foods12234296_

Round 1

Reviewer 1 Report

Comments and Suggestions for Authors

Please add data of the non-ensiled forages i.e. on day 0. This is very critical to show the chemical composition of the forage before ensiling. We want to know how much WSC can be expected in the forage before ensiling.

Comments on the Quality of English Language

Line 33: replace developed with bred

L81: replace was with were

L251-274: should be should be summarized 

L287-288: rephrase to make sense

L294: should read "the pH of silage is an essential indicator for good fermentation"

L372: replace developed with produced 

Author Response

We thank the reviewers for their critical and judicious evaluation of our manuscript, and for providing constructive suggestions for improving its quality. All reviewers' comments have been carefully considered and the manuscript has been thoroughly revised. I have responded to the reviewer's comments point by point. Red fonts were used for all changes in manuscript.

  1. Line 33: replace developed with bred

A triticale crop was developed in German laboratories from wheat and rye in the 19th century.

  1. L81: replace was with were

The manuscript has been edited by MDPI, a language editing service

  1. L251-274: should be should be summarized 

Thank you for your kind suggestion and revised as There is evidence that the LAB treatments performed in mixed form produced higher LA and significantly reduced unwanted microbes in comparison with single strain treatments. In addition, LAB co-culture treatment extends the fermented silage storage time while preserving nutrients

  1. L287-288: rephrase to make sense

The language of the manuscript has been edited by MDPI language editing service

  1. L294: should read "the pH of silage is an essential indicator for good fermentation"

The following sentences have been revised as In order to produce high-quality silage, the pH is an essential factor. When the pH is rapidly reduced in silage, the proteolytic enzyme activity is reduced, preventing nutrient decomposition and inhibiting the growth of unwanted bacteria, including enterobacteria and clostridia

  1. L372: replace developed with produced

Thank you for your information and revised as Silage produced with different LAB, either as a single bacterium or mixed LAB, significantly reduced the pH and butyric contents and increased lactic acid contents by inhibiting the growth of yeast, mold and other unwanted bacteria

Reviewer 2 Report

Comments and Suggestions for Authors

This study investigates the impact of ensiling triticale with Lactic Acid Bacteria (LAB) on anaerobic fermentation and the resultant quality of grass-based animal feed. The findings provide valuable insights into the benefits of using LAB in enhancing the nutritional quality of silage and reducing unwanted microbial growth. Here are some key points to consider in the general evaluation.

Please unify the time unit in Line 82 and 95. The entire text should have a space between numbers and units. Moreover, the results in Section 3.4 lacks complete unit specifications for quantities. Please provide the full unit information for clarity in this section.

The tables in this manuscript are of poor quality, with numerous detailed errors that require correction. The authors can reformat Table 3 to match the style of Tables 1 and 2. Additionally, to enhance the readability of the article, consider converting some of the data that warrant discussion into graphical representations for a more intuitive presentation. These tabular data can be submitted as supplementary materials for the benefit of the readers.

The results analysis in Table 6 appears to be a duplication of the content from Table 5. This is a significant and unacceptable error that needs to be rectified.

Please reconfirm the LAB count for the control group at low moisture in Table 5, and ensure the accuracy of the results in Table 5 and Table 6.

Comments on the Quality of English Language

There are some English grammar issues in this manuscript, including errors in singular and plural usage (Line 81). Authors should review the grammar throughout the entire text. Good grammar and language expression are crucial for the readability and professionalism of scientific research articles.

Author Response

At the outset we thank the reviewers for their critical and judicious evaluation of our manuscript, and providing constructive suggestions for improving the quality and presentation of the manuscript. We have carefully considered the comments of the reviewers and revised the manuscript thoroughly taking all the points into consideration. Point wise response to the reviewer's comments is given below. All changes in manuscript were made with red color fonts.

This study investigates the impact of ensiling triticale with Lactic Acid Bacteria (LAB) on anaerobic fermentation and the resultant quality of grass-based animal feed. The findings provide valuable insights into the benefits of using LAB in enhancing the nutritional quality of silage and reducing unwanted microbial growth. Here are some key points to consider in the general evaluation.

. Thank you for your four valuable suggestions and positive comments regarding the manuscript submitted

  1. lease unify the time unit in Line 82 and 95. The entire text should have a space between numbers and units. Moreover, the results in Section 3.4 lacks complete unit specifications for quantities. Please provide the full unit information for clarity in this section.

Thank you very much for your valuable suggestion. Throughout the manuscript, we have unified all units, revised the quantities specifications in section 3.4, and explained CFU/g in figure legends.

Microbial populations, such as the total lactic acid bacteria (LAB), yeast and mold, were counted in both HM and LM silages on day 180 in response to different LAB treatments (Figure 3a-c). Under high moisture conditions, the number of LAB was 11.5 x 6.3 CFU/g, while under low moisture conditions, it was 05.0 ± 1.4 CFU/g in the control. The yeast and mold counts for control HM silage were 341 ± 1.55 and 20.5 ± 6.36 CFU/g and for LM silage they were 24.3 ± 0.40 and 12.2 ± 1.25 CFU/g, respectively. Silage treated with different types of LAB, either as a single or co-culture, had higher LAB populations than the control silage. The LAB numbers in HM silage varied with inoculant treatments, ranging from 30.5 ± 4.9 to 46.5 ± 7.7 CFU/g. The yeast and mold counts were also drastically reduced or insignificant when silage was treated with different LAB monocultures or co-cultures.

  1. The tables in this manuscript are of poor quality, with numerous detailed errors that require correction. The authors can reformat Table 3 to match the style of Tables 1 and 2. Additionally, to enhance the readability of the article, consider converting some of the data that warrant discussion into graphical representations for a more intuitive presentation. These tabular data can be submitted as supplementary materials for the benefit of the readers.

We strongly agree with the reviewer's comments and suggestions, so most tables have been converted to images in the manuscript. We hope that readers will be able to better understand the revised figure.

  1. the results analysis in Table 6 appears to be a duplication of the content from Table 5. This is a significant and unacceptable error that needs to be rectified.

Thank you for sharing your information. Tables 5 and 6 have been modified into graphical representations with relevant information. Tables 5 and 6 represent the microbial population in experimental silage on days 180 and 360, respectively. It was unclear to me where the duplicate content could be found in the tables. However, the whole table and its legends have been carefully checked and revised

  1. lease reconfirm the LAB count for the control group at low moisture in Table 5, and ensure the accuracy of the results in Table 5 and Table 6.

Thank you for providing this information. Tables 5 and 6 have been converted into graphical representations with relevant information. In order to achieve uniformity, we used 05.0 ± 1.4 instead of 5.0 ± 1.4

  1. There are some English grammar issues in this manuscript, including errors in singular and plural usage (Line 81). Authors should review the grammar throughout the entire text. Good grammar and language expression are crucial for the readability and professionalism of scientific research articles.

Thank you for taking the time to comment on the manuscripts' language. The manuscript has been edited by MDPI language editing service (English editing ID: English-73932).

Reviewer 3 Report

Comments and Suggestions for Authors

Ms is written nicely and can be accepted in the present form.

Author Response

The manuscript is written nicely and can be accepted in the present form

Thank you for your positive comments and recommendation to accept the manuscript submitted

Reviewer 4 Report

Comments and Suggestions for Authors

This manuscript by Soundharrajan et al. investigated the effect of lactic acid bacteria to the anaerobic fermentation in triticale silage. This manuscript generally fits the scope of Foods journal. I have some comments which need revision by the authors to the manuscript.

1. In Materials and Methods, section 2.3, the conditions used for HPLC and GC can be added to provide more details.

2. The tables in this manuscript can be changed to figures such as column graphs. The description of these tables in the results section can also be revised using the graphs. The titles for these tables can be revised as figure legends. 

Author Response

This manuscript by Soundharrajan et al. investigated the effect of lactic acid bacteria to the anaerobic fermentation in triticale silage. This manuscript generally fits the scope of Foods journal. I have some comments which need revision by the authors to the manuscript.

We thank the reviewers for their careful and constructive evaluation of our manuscript, as well as for providing constructive suggestions on how to improve it. All the points raised by the reviewers have been carefully considered and revised. Below is a point by point response to the reviewer's comments. Changes in manuscript were made in red font.

  1. In Materials and Methods, section 2.3, the conditions used for HPLC and GC can be added to provide more details.

A detailed description of HPLC and GC analysis protocol and conditions has been added to the method section in response to a reviewer's suggestion. To analyze fermentation characteristics, 10 grams of each sample was mixed with 90mL of water and shaken in an orbital shaker for an hour. A glass electrode pH meter (Thomas Scientific, NJ, USA) was used to measure the pH of the filtrate after it had been passed through multiple layers of cheesecloth and a 0. 2 μm filter membrane. Afterwards, organic acid concentrations in the fermented samples were determined. Samples were reduced to a pH of 2 with 50% sulfuric acid and frozen at -20°C for HPLC analyses. The lactic acid concentration in experimental silage was determined using a high-performance liquid chromatography system (HP1100, Agilent Co., USA). A Hi-Plex Ligand exchange column (300 x 7.7mm) from Agilent was used to elute the sample at 40°C with 0.1M H2SO4. HPLC analysis was conducted at a flow rate of 0.6mL/min at a wavelength of 220 nm. CP7485 column fused with silica (Length- 25cm, diameter- 0.32, film thickness= 0.30), temperature ranges from 20°C to 270°C, was used to analyze acetic and butyric acid content in silage. The sample flow rate was 10 microliters per minute [3,23].

  1. The tables in this manuscript can be changed to figures such as column graphs. The description of these tables in the results section can also be revised using the graphs. The titles for these tables can be revised as figure legend.

We strongly agree with the reviewer's comments and suggestions, so most tables have been converted to images in the manuscript. We hope that readers will be able to better understand the revised figure.

Reviewer 5 Report

Comments and Suggestions for Authors

An interesting study, however some issues should be improved:

1. please check spaces, lines 52-60 missing space before []

2. please add spaces between numerical values and units in methods section

3. Line 104 - please provide details for LAB growth conditions - aerobically/anaerobically/static/shaking?

4. line 109 - 0.2 m?

5. line 128 - please provide details for LAB growth condidtions

6. Lines 161-175, etc. - please use and instead of &

7. Line 274 - please remove on dot (..)

8. What was the status of antinutritive or toxic compounds (mycotoxins)?

Comments on the Quality of English Language

minor isssues, please double check

Author Response

An interesting study, however some issues should be improved:

Thank you to the reviewers for providing constructive suggestions on how to improve our manuscript. Revisions have been made to all points raised by reviewers. Here is a point by point response to the reviewer's comments. Changes in manuscript are in red.

  1. please check spaces, lines 52-60 missing space before []

Thank you for your kind information. We have checked whole manuscript and removed or added spaces between words. 

  1. please add spaces between numerical values and units in methods section

Thank you for your kind information. We have checked whole manuscript and removed or added spaces between words and numerical values. 

  1. Line 104 - please provide details for LAB growth conditions - aerobically/anaerobically/static/shaking?

Thank you for your suggestions. The details of LAB growth conditions were updated in methods section as follows. All strains were cultured in MRS broth (CONDA, Madrid, Spain) for 30h with mild shaking at 150rpm in an orbital shaker under micro-aerobic conditions at 37°C and the pellets were removed by centrifugation at 4000g for 45 minutes at 4°C.

  1. line 109 - 0.2 m?

It is purely typographical errors which has revised now as A glass electrode pH meter (Thomas Scientific, NJ, USA) was used to measure the pH of the filtrate after it had been passed through multiple layers of cheesecloth and a 0. 2 μm filter membrane

  1. line 128 - please provide details for LAB growth conditions

A detail of the LAB, yeast and mold counts was described in method section as  For enumeration of, LAB, molds, and yeast, a portion of sample was filtered with sterilized cheesecloth and then serially diluted tenfold in sterile distilled water, 0.1 mL each sample was poured on de Man Rogosa Sharpe Agar plate (MRS agar, CONDA, Madrid, Spain) and incubated at 37°C for 48 hours under aerobic condition. Further, 1 mL diluted sample was spread on petriflim for detection of molds, and yeast (3M microbiology products, St. Paul, USA) and incubated at 37°C, for 120. After respective incubation periods, the population of microbes were enumerated. The population of microbes was enumerated after the respective incubation periods.

  1. Lines 161-175, etc. - please use and instead of &

The sentences between lines 161 and 175 have been revised during the language edition process.

  1. Line 274 - please remove on dot (..)

The extra dot was removed throughout the manuscript in response to a reviewer's suggestion.

  1. What was the status of antinutritive or toxic compounds (mycotoxins)?

Please accept my sincere thanks for your question. It does not belong to the mycotoxin family. Forages contain antinutritionals factors that disturb normal biological processes, such as degradation of ruminal microflora, reduced absorption, etc. The compounds are trypsin inhibitors, phytic acid, glycosides, alkaloids, triterpenes, oxalates, polyphenols, and condensed tannins.

,

Round 2

Reviewer 5 Report

Comments and Suggestions for Authors

The authors have adressed all issues.

Author Response

Thank you for sending corrections and suggestions to our manuscript, and for your prompt response as well.

In this email, I have attached a revised manuscript with a response to your comments.